# The Origins of the Dark—Hyperactivity and Negative Peer Relationships, an Objectively Lower Sleep Efficiency, and a Longer Sleep Onset Latency at Age Five Were Associated with Callous-Unemotional Traits and Low Empathy at Age 14

**DOI:** 10.3390/jcm12062248

**Published:** 2023-03-14

**Authors:** Larina Eisenhut, Dena Sadeghi-Bahmani, Vinh Tong Ngo, Thorsten Mikoteit, Annette Beatrix Brühl, Christina Stadler, Kenneth M. Dürsteler, Martin Hatzinger, Serge Brand

**Affiliations:** 1Center for Affective, Stress and Sleep Disorders (ZASS), Psychiatric University Hospital Basel (UPK), 4002 Basel, Switzerland; 2Department of Psychology, Stanford University, Stanford, CA 94305, USA; 3Sleep Disorders Research Center, Kermanshah University of Medical Sciences (KUMS), Kermanshah 6714869914, Iran; 4Centre for Addictive Disorders, Department of Psychiatry, Psychotherapy and Psychosomatics, Psychiatric Hospital, University of Zurich, 8001 Zurich, Switzerland; 5Psychiatric Services Solothurn, University of Basel, 4503 Solothurn, Switzerland; 6Child and Adolescent Research Department, Psychiatric University Hospital Basel (UPK), 4002 Basel, Switzerland; 7Division of Substance Use Disorders, Psychiatric University Hospital Basel, 4002 Basel, Switzerland; 8Substance Abuse Prevention Research Center, Kermanshah University of Medical Sciences (KUMS), Kermanshah 6714869914, Iran; 9Division of Sport Science and Psychosocial Health, Department of Sport, Exercise and Health, University of Basel, 4002 Basel, Switzerland; 10School of Medicine, Teheran University of Medical Sciences, Teheran 1417466191, Iran; 11Center for Disaster Psychiatry and Disaster Psychology, Psychiatric University Hospital Basel (UPK), 4002 Basel, Switzerland

**Keywords:** callous-unemotional traits, empathy, longitudinal study, objective sleep

## Abstract

Background: Within the spectrum of emotional competencies, callous-unemotional traits are socially discouraged, while empathy is considered a socially much more accepted emotional trait. This holds particularly true for adolescents, who are still building up their social and emotional competencies. The aims of the present study were two-fold: First, longitudinally, to identify traits of behavioral problems and objective sleep dimensions at the age of 5 years to predict callous-unemotional traits and empathy at the age of 14 years. Second, cross-sectionally, to associate callous-unemotional traits and empathy with current insomnia, stress, and mental toughness. Methods: Preschoolers at the age of 5 years were contacted nine years later at the age of 14 years. At 5 years, parents rated their children’s behavior (Strength and Difficulties Questionnaire, SDQ); in parallel, children underwent a one-night sleep-EEG assessment. At the age of 14 years, adolescents completed a series of questionnaires covering callous-unemotional traits, insomnia, empathy, stress, and mental toughness. Results: A total of 77 adolescents (38.1% females) took part in the present study. Longitudinally, higher scores for hyperactivity at age 5 significantly predicted higher callous-unemotional traits at age 14. A higher score for negative peer relationships at age 5 significantly predicted lower scores for cognitive empathy at age 14. Further, objective sleep-EEG measures showed that a higher sleep efficiency and a shorter sleep latency was associated with lower scores for callousness. Cross-sectionally, higher scores for callous-unemotional traits were associated with higher insomnia and stress, while lower insomnia was associated with higher empathy. Mental toughness was unrelated to callous-unemotional traits and empathy. Conclusions: It appears that hyperactivity traits and negative peer relationships and more unfavorable objective sleep patterns at 5 years predicted socially discouraged callous-unemotional traits and low empathy during adolescence. Further, cross-sectionally at the age of 14, callous-unemotional traits, subjective poor sleep, and higher stress were associated.

## 1. Introduction

From the perspective of developmental psychology, behavior and experiences in childhood might have longer-lasting effects later in life [1,2,3,4]. Further, about 50% of all psychiatric diagnoses emerge during the first 15 years of life [5,6]. As such, psychological and behavioral problems and psychiatric issues in childhood and adolescence are common. Importantly, former methodologically well-conducted longitudinal studies showed that childhood behavior and experiences did not linearly lead to psychological issues in later life [7,8]. However, there is increasing evidence that childhood emotional and behavioral problems are associated with a wide range of adverse outcomes in adulthood [9,10,11]. In the present study, we focused on emotional and behavioral problems in adolescence as the possible result of emotional and behavioral patterns and objective sleep dimensions at the age of 5 years. 

### 1.1. Callous-Unemotional Traits

Callous-unemotional traits are a dimensional construct that encompasses a symptom subset of empathy impairment and affective emotions [12]. Children and adolescents with callous-unemotional traits are characterized by a lack of empathy, a sense of guilt, and shallow emotions [13,14,15]. There is evidence that a lack of empathy could be considered a core feature of callous-unemotional traits in children and adolescents [16,17], and characterizing these traits in adolescents may improve the diagnostic, prognostic, and therapeutic possibilities [17]. Further, low interpersonal emotional sensitivity and fearlessness are thought to play an important role in callous-unemotional traits [18,19]. Longitudinally, there is evidence that fearlessness at the age of 2–3 years is linked to callous-unemotional traits in adolescence [20] and psychopathy in adults [21].

Callous-unemotional traits are reinforced by environmental risk factors such as strict parenting practices and a lack of parental warmth [19]. Additonally, it appears that children and adolescents with callous-unemotional traits have specific genetic, cognitive, emotional, biological, and environmental characteristics, some of which are similar to those of adults with psychopathy [22,23]. Given this, the first aim of the present study was to investigate whether emotional and behavioral patterns in childhood are related to callous-unemotional traits in adolescence.

### 1.2. Empathy

Empathy is the ability to share and understand the emotional states of others and has major impact on how people act in social situations [24]. Empathy is considered a multidimensional construct, encompassing cognitive, affective, and behavioral aspects [17], though empathy is usually conceptualized in two dimensions: affective and cognitive empathy. Affective empathy refers to the ability to share the emotions of others, while cognitive empathy involves the ability to infer or understand the emotional experiences of others [25].

The development of empathy is considered an essential part of moral development and plays an important role in the development of social competence [26,27]. Deficits in empathy during childhood appear to contribute to psychopathology in later life and appear to be associated with antisocial behavior, bullying, aggression, and crime [27,28,29]. Understanding the process involved in the development of empathy may help to develop interventions to reduce aggressive and antisocial behavior, particularly among individuals at risk for such behaviors, such as adolescents with callous-unemotional traits [19]. There is already research examining the development of empathy from childhood into adulthood [30,31]. However, the relationship between behavioral problems in childhood and empathy in adolescence has been less studied. The second aim of the present study was therefore to investigate whether there is a relationship between early behavioral problems and later empathy impairments.

### 1.3. Psychiatric Issues from Childhood to Adolescence and Young Adulthood

There is evidence that symptoms of psychopathy in preschoolers can persist over time, and that psychiatric issues in adolescence can be predicted by psychopathology in childhood [5,6,32,33]. In particular, the extent to which behavioral problems such as internalizing and externalizing behaviors in preschool children can predict psychiatric problems in adolescence and adulthood has been studied. However, externalizing problems have been studied more frequently than internalizing problems. Results to date suggest greater stability over time for externalizing than for internalizing behavior problems [34]. In addition, there is evidence that externalizing problems in boys and girls at age three predict both externalizing and internalizing problems at age 12 [34].

Externalizing behaviors are characterized by a variety of antisocial behaviors, such as verbal and physical harm to others; the violation of social expectations; and behaviors such as delinquency, vandalism, and theft, to name but a few [35,36]. Externalizing disorders in children and adolescents include conduct disorder (CD), oppositional defiant disorder (ODD), and attention deficit hyperactivity disorder (ADHD) [37]. Further, externalizing behaviors are associated with social aggression, disruptive behavior, and risky behaviors [34]. In general, children with early onset behavioral problems that begin in preschool seem to have a more negative prognosis than children with a later onset only in adolescence [38]. In summary, psychopathologies such as externalizing problems in childhood and adolescence seem to predict unfavorable behaviors such as tobacco use later in life [39].

As for internalizing problems, typical traits consist of anxious and depressive symptoms, withdrawn behavior, and somatic symptoms [40]. Goodwin et al. [41]. reported that internalizing problems in childhood were associated with anxiety and depression in adolescence and young adulthood.

Remarkably, the current literature consists predominantly of studies that focused on the effects of internalizing or externalizing problems on maladaptive behavior in adulthood, ignoring their co-occurrence. However, Waller and Hyde [18] found that boys with high scores for internalizing and externalizing problems at age 8 were at risk for psychiatric disorders, offending, and substance abuse in adulthood.

Regarding sleep problems over time, Wong et al. (2010) [42] reported that sleep problems at ages 3 to 8 were significantly related to self-reported sleep problems at ages 11 to 17. Brand et al. [43] further found that children with sleep problems in childhood were almost 2.5 times more likely to have sleep problems in adolescence compared to children without sleep problems. In addition, sleep and internalizing and externalizing problems in childhood and early adulthood have been found to be related to each other [44,45,46,47,48].

### 1.4. Callous-Unemotional Traits, Empathy, and Sleep Patterns

Good sleep is essential for various aspects of health and psychological functioning [49]. The long-term consequences of sleep disruption in otherwise healthy individuals include hypertension, cardiovascular disease, weight-related issues, metabolic syndrome, and type 2 diabetes mellitus [50]. Recent research even confirmed an association between sleep disorders and psychopathy [51]. Akram et al. [52] reported an association between sleep disturbances and individuals with high psychopathy in terms of deficits in emotion regulation. These deficits in emotion regulation served to accentuate negatively toned cognitive activities, which in turn led to poor sleep [52]. Research regarding psychopathy and sleep is surprisingly deficient. However, as far as callous-unemotional traits are concerned, the body of studies is even more sparse and currently inconclusive. Denis, et al. [53] performed two separate studies: In the first study, among 1556 young adults aged 18–27 years (62% females), no significant associations were observed between subjectively rated sleep quality and callous-unemotional traits [53]. In the second study, among 43 adults aged 18–66 years (65% females), better actigraphically measured sleep efficiency was associated with higher scores for callous-unemotional traits [53]. Next, among 239 clinic-referred youth aged 6 to 17 years, callous-unemotional traits were unrelated to their sleep problems [54]. Given this, it appears that callous-unemotional traits are either unrelated or even favorably related to better sleep patterns.

As regards the associations between sleep patterns and empathy, in a previous study [55] we showed that adolescents scoring low in subjective sleep quality also scored low in empathy traits; as such, it appeared plausible that empathy and restoring sleep are associated.

Overall, the pool of studies on this specific research area is sparse, and their results appear mixed. We took these observations into account and investigated whether callous-unemotional traits and empathy might be associated with subjectively assessed sleep patterns among adolescents.

### 1.5. Callous-Unemotional Traits, Empathy, and Stress

The expression of callous-unemotional traits is considered to be a product of a transactional cycle of hypoarousal in peripheral stress systems and limbic circuitry [56]. Since low cortisol levels can reduce emotional or stress responses, even in social contexts where stress responses are appropriate [57], Shirtcliff et al. [58] pointed to an important contribution of the HPA axis to the development of callous-unemotional traits. Del Giudice et al. [59] held that the insufficient mobilization of resources reduces the ability of the stress response system to monitor and encode environmental threats, leaving the organism vulnerable to further physiological impairment in the face of future stressors.

However, few studies have investigated the relationship between callous-unemotional traits and stress.

Gostisha et al. [56] examined whether psychopathic symptoms or life stress exposure are associated with cortisol and its diurnal rhythm within incarcerated adolescent boys. It was found that steeper cortisol awakening response increases were associated with lifetime stress exposure and callous-unemotional traits. There is also evidence that callous-unemotional traits were negatively associated with stress response [60]. Regarding empathy, a previous study showed a negative correlation between empathy and stress in medical students [61].

In summary, there is scant evidence of the relationship between callous-unemotional traits, empathy, and stress, and the current data are very sparse. For this reason, we investigated whether callous-unemotional traits and empathy might be related to adolescents’ subjective perceived stress.

### 1.6. The Current Study

Previous research has focused on the predictive value of internalizing and externalizing problems and sleep disturbances in childhood for psychopathological traits in adolescence, but not for callous-unemotional traits and empathy. Moreover, to our knowledge, no study has yet investigated which psychological dimensions in childhood are correlated with and might predict callous-unemotional traits and empathy in adolescence. The aim of the present study was therefore to investigate these research questions. To this end, participants of a previous study (age five) were contacted again nine years later and thus assessed at the age of 14.

Given the lack of previous research, we drew on findings from cross-sectional studies to formulate the following hypotheses. First, following other studies, we expected that positive psychological traits in childhood would be associated with and even predict lower callous-unemotional traits [53,62,63] and higher empathy scores [64,65,66] at age 14 (Hypothesis 1). 

Next, based on a previous study [67], we expected that higher sleep quality longitudinally (objective measure of sleep parameters at age five; Hypothesis 2) and lower scores for insomnia cross-sectionally (subjective measure of insomnia at age 14; Hypothesis 3) would be associated with lower callous-unemotional traits at age 14. Similarly, with the fourth and fifth hypotheses, we predicted that a higher sleep quality longitudinally (objective measure of sleep parameters at age five; Hypothesis 4) and lower scores for insomnia cross-sectionally (subjective measure of insomnia at age 14; Hypothesis 5) would be associated with higher scores for empathy at age 14. With the sixth hypothesis we expected that higher scores for callous-unemotional traits and lower scores for empathy cross-sectionally would be associated with higher perceived stress and lower mental toughness traits.

We hold that the present results have the potential to add to the current literature in the following three important ways. First, we filled the research gap regarding the predictive value of objective sleep measures during the preschool stage and callous-unemotional traits during adolescence; second, relatedly, we filled the research gap regarding the predictive value of behavioral traits during the preschool stage and callous-unemotional traits and empathy during adolescence; and third, we investigated whether callous-unemotional traits and empathy were associated with current psychological concerns related to stress and insomnia and with mental toughness.

## 2. Materials and Methods

### 2.1. Participants

A detailed description of the core sample has already been provided elsewhere and can be found in Brand et al. [43] or Perren et al. [68]. In brief, preschool children from 18 different kindergartens in Basel (the northwestern and German-speaking part of Switzerland) were assessed. Those 18 kindergartens were randomly chosen, though considering a balanced stratification of different quartiers reflecting a broad range of socioeconomic backgrounds, including prevalence rates of immigrants with (Swiss-) German as a second language. Eligible preschoolers were assessed for their subjective and objective sleep parameters and under challenge conditions (a modified social stress test), and psychological functioning (Strength and Difficulties Questionnaire (SDQ), see description of the measure below: preschoolers’ strengths and difficulties—parent rating) as assessed via parents’ and teachers’ ratings. Approximately 9 years later, 84 (88.43%) of the 95 children aged five years (M = 5.4 years, SD = 0.44) for whom parents completed the SDQ agreed to participate in the follow-up study at age 14 years. At follow-up, the mean age of the adolescents was 14.25 years (SD = 1.21; 32 females and 45 males). Please note that while cross-sectional data were available for 84 participants, longitudinal data were available for 77 participants. None of the participants reported any kind of sleep or mood medications. As already shown [43], participants and non-participants did not differ significantly at follow-up with regard to age, gender, sleep profiles, or psychological traits at baseline. Accordingly, age, gender, sleep profiles, and psychological traits were not included as covariates in all statistical equations of the present study.

### 2.2. Procedure

As described elsewhere [43], the participating children were examined in their first year of kindergarten (at the age of 5; [69,70]). The same children were contacted and examined again at the age of 14. Participants completed self-assessment questionnaires on socio-demographic data, sleep, callous-unemotional behavior, and empathy (see description below: callous-unemotional traits). The adolescents and their parents were informed about the purpose of the follow-up study. Both adolescents and parents were required to sign an informed consent form before participating in the study. The study protocol was conducted in accordance with the Declaration of Helsinki and approved by the local ethics committee. Parts of this longitudinal study have already been published. Brand et al. [43] showed that sleep quality at age five predicted psychological characteristics in areas such as peer relationships and success in coping with stress. Sadeghi Bahmani et al. [71] showed that higher prosocial behavior, lower negative peer relationships, and lower internalizing problems and externalizing problems at age five, rated by parents and teachers, were associated with higher self-rated mental toughness and lower sleep disturbance at age 14. In the present study, we focused on the associations between psychological characteristics and sleep quality at age five (SDQ, parent ratings, and EEG data; see description below: preschoolers’ objective sleep patterns) and self-rated callous-unemotional behavior and empathy (ICU and GEM; see below: callous-unemotional traits; empathy). This pattern of correlations has not been studied before. The present data are therefore new.

### 2.3. Measures

#### 2.3.1. Measures Applied at the Age of Five Years

##### Preschoolers’ Strengths and Difficulties—Parent Rating

To rate preschoolers’ psychological and behavioral strengths and difficulties, parents completed the Strengths and Difficulties Questionnaire (SDQ; [72]). The questionnaire consists of 25 items covering the following dimensions: internalizing problems, externalizing problems, hyperactivity, peer problems, and prosocial behavior. In addition, a total score can be obtained, with a higher score reflecting more negative psychological functioning. Each subscale consists of 5 items that are rated on a three-point scale (0 = strongly disagree, 2 = strongly agree). The sum is calculated to determine the scores of the subscales. The SDQ is considered a valid behavioral screening tool in RCTs and clinical settings [73], both in children and adolescents ([74,75]). In the present study, the internal consistency was moderate to high (Cronbach’s alpha = 0.87).

##### Preschoolers’ Objective Sleep Patterns

As described in detail elsewhere [69,76,77], the sleep of the preschool children was recorded with the ambulatory EEG system (Oxford Medilog) during one night at the children’s homes. The first night was used to familiarize the children with the recording and the attachment of the electrodes. The electroencephalographic registration took place on the second night. Electrodes for polysomnographic recordings (electroencephalogram: C3-A2, C4-A1; electromyogram: two electrodes on the chin; electrooculogram: two electrodes on the right and left side) were fixed the evening before registration and removed the next morning. Sleep polygraphs were visually analyzed by two experienced raters according to standard procedures [78].

#### 2.3.2. Measures Applied at the Age of 14 Years

##### Callous-Unemotional Traits

To self-assess callous-unemotional traits, adolescents completed the Inventory of Callous-Unemotional Traits (ICU; [79]). The ICU is the most commonly used measure of callous-unemotional traits and is used in self-report and observational report versions [80,81]. For the present study, the self-report version was used. The ICU consists of 24 items from which three subscales can be calculated: callous, uncaring, and unemotional [80]. Each item is rated using a four-point rating scale (0 = not at all true, 3 = definitely true), with a higher score reflecting more callous-unemotional behavior. In addition, a total score can be calculated. The ICU total score (Frick et al., 2014; Frick and Ray 2015) [14,18], as well as the callous and uncaring subscales (Cardinale and Marsh 2020) [80] are considered to be a valid measure for callous-unemotional traits in children and adolescents. However, Cardinale and Marsh (2020) [80] also pointed out the weakness of the unemotional subscale as it relates to interpersonal callousness, uncaring, and antisociality. The total ICU scores are considered to be a valid continuous measure of callous-unemotional traits [14,18] (Cronbach’s alpha = 0.85).

##### Empathy

To assess empathy traits, adolescents completed the Griffith Empathy Measure (GEM; [82]). The GEM consists of 23 items that are answered on a nine-point rating scale (−4 = strongly disagree, 4 = strongly agree). The GEM can be used as a single scale based on the overall score of the items; alternatively, after omitting items that load on both subscales, it can be scored in cognitive and affective subscales. A higher score reflects greater empathic behavior. The GEM can be used in children and adolescents to measure empathic behavior and shows good reliability and validity across gender and age [82]. (Cronbach’s alpha = 0.90.)

##### Insomnia

To assess insomnia, participants completed the Insomnia Severity Index (ISI; [83]). The 7-item questionnaire is a comprehensive screening measure for insomnia. Items are answered using a five-point rating scale (0 = not at all, 4 = very much) and measure difficulty falling asleep, difficulty staying asleep, early morning awakenings, increased daytime sleepiness, impaired daytime sleepiness, impaired daytime performance, low satisfaction with sleep, and worry about sleep. The validity and reliability of the ISI has been proven previously ([84,85]). The higher the total score, the more the respondent is assumed to suffer from sleep disturbances (Cronbach’s alpha = 0.92).

##### Perceived Stress

To assess self-perceived stress, adolescents completed the Perceived Stress Scale (PSS; [86]). The items of the questionnaire capture how unpredictable, uncontrollable, and overloaded respondents perceive their lives to be [87]. The questionnaire consists of 10 items from which the total stress experienced during the previous month can be determined [86]. Responses are provided on a five-point rating scale, with higher scores reflecting greater perceived stress. Various studies have shown that the PSS is a valid instrument for assessing perceived stress in different languages, cultures, and age groups [88,89,90,91]. (Cronbach’s alpha = 0.90).

##### Mental Toughness

To assess mental toughness, participants completed the German version of the 18-item Mental Toughness Questionnaire (MTQ18; [92]. The MTQ18 is a shortened version of the MTQ48 questionnaire. The MTQ48 has been shown to be a valid and reliable instrument in previous research [93,94]. Further, research showed that there are very high correlations between the MTQ18 and the MTQ48. The answers are provided on a five-point rating scale, ranging from 1 (strongly disagree) to 5 (strongly agree). For the evaluation, the individual items were added to a total score, with a higher score reflecting a greater mental toughness (Cronbach’s alpha = 0.92).

### 2.4. Statistical Analysis

For longitudinal analyses (n = 77), and to answer to Hypotheses 1, 2, and 4, first, a series of Pearson’s correlations was performed between parents’ assessments of children’s strengths and difficulties at age five (SDQ), objective sleep parameters (EEG-recordings at age five), and participants’ ratings at the age of 14: callous-unemotional traits and empathy.

Second, and again to answer to Hypotheses 1, 2, and 4, three multiple regression analyses were performed with callous-unemotional traits, callousness, and empathy (age 14) as dependent variables and SDQ scores and sleep-EEG-dimensions (age five) as predictors. The preliminary conditions to perform multiple regression analyses were generally met [95,96,97]: predictors explained the dependent variables (Rs = 0.392–0.394, R^2^s = 0.153–0.155); the number of predictors was 7; 10 × 7 = 70 < N (77); and the Durbin–Watson coefficients were 1.914–2.321, indicating that the residuals of the predictors were independent. Furthermore, the variance inflation factors (VIFs) were between 1.151 and 1.190, and while there are no strict cut-off points to report the risk of multicollinearity, VIF < 1 and VIF > 10 indicate multicollinearity [97,98].

Third, to answer to hypotheses 3, 5, and 6, for cross-sectional analyses at the age of 14 (n = 81–84), a series of Pearson’s correlations was performed between callous-unemotional traits, empathy, insomnia, stress, and mental toughness. 

The nominal significance level was set at alpha < 0.05. Statistical procedures were performed using SPSS^®^ 27.0 (IBM Corporation, Armonk, NY, USA) for Apple Mac^®^.

## 3. Results

### 3.1. Sociodemographic Information

At follow-up, the mean age of the adolescents was 14.25 years (SD = 1.21; 32 females and 45 males). Please note that while cross-sectional data were available for 84 participants, longitudinal data were available for 77 participants.

### 3.2. Longitudinal Results

#### 3.2.1. Associations between Strengths and Difficulties and Objective Sleep Patterns at the Age of Five Years and Callous-Unemotional Traits and Empathy at the Age of 14 Years

Descriptive statistics and bivariate correlations between strengths and difficulties (SDQ) and objective sleep (sleep-EEG) at five years and callous-unemotional traits (ICU) and empathy (GEM) at 14 years are reported in Table 1.

More externalizing problems as rated by parents in childhood were correlated with more callousness towards others and feelings (ICU callousness) and higher callous-unemotional traits in terms of the overall score (ICU overall score). Increased hyperactivity as rated by parents was associated with more callousness towards others and feelings (ICU callousness), more indifference to one’s own performance and actions (ICU uncaring), more emotional closure (ICU unemotional), and more callous-unemotional traits in terms of the overall score (ICU overall score) at age 14. Furthermore, children scoring high in the overall score showed more indifference to their own performance and actions (ICU uncaring), as well as more callous-unemotional traits in terms of the overall score (ICU overall score). Internalizing problems, negative peer relationships and prosocial behavior were not correlated with callous-unemotional traits.

More externalizing problems, more negative peer relationships, and higher overall scores as rated by parents in childhood were associated with reduced cognitive empathy. Better prosocial behavior as rated by parents was correlated with higher overall scores in empathy.

In terms of objective sleep, a better sleep efficiency at the age of five years was associated with lower callousness scores at the age of 14. In contrast, a longer sleep latency at the age of five years correlated with more callousness (ICU callousness) at the age of 14. Time in bed (TIB) and total sleep time (TST) at the age of five years were not related to callous-unemotional traits at the age of 14.

Further, a longer sleep latency at the age of five years was also associated with higher cognitive empathy at the age of 14. Otherwise, no correlations were observed between objective sleep and empathy.

#### 3.2.2. Strengths and Difficulties and Objective Sleep Patterns at the Age of Five to Predict Callous-Unemotional Traits, Callousness, and Empathy at the Age of 14

Table 2 shows the results of the two multiple regression analyses with the callous-unemotional overall score and cognitive empathy as dependent variables and the strengths and difficulties variables (to avoid redundancy and biased calculations, the SDQ totals were not included in the equations), objective sleep efficiency, and objective sleep latency as predictors.

Higher scores for hyperactivity at age five significantly predicted higher callous-unemotional traits at age 14. Further, a lower score for negative peer relationships, that is, more favorable peer relationships, at age five significantly predicted higher scores for cognitive empathy at age 14.

### 3.3. Cross-Sectional Results

#### Callous-Unemotional Traits, Empathy, Insomnia, Stress, and Mental Toughness at the Age of 14

Descriptive statistics and bivariate correlations between callous-unemotional traits and empathy and insomnia (ISI), perceived stress (PSS), and mental toughness (MT) are shown in Table 3.

Higher scores for insomnia were associated with increased callousness (ICU callousness), increased emotional closure (ICU unemotional), and higher overall scores for callous-unemotional traits. Better coping with stress and mental toughness (MT) were associated with less emotional closure. Furthermore, higher perceived stress (PSS) was also associated with more indifference to one’s own performance and actions (ICU uncaring). 

For empathy traits, lower scores for cognitive empathy were associated with higher scores for insomnia. No statistically significant correlations were observed between empathy traits and perceived stress and mental toughness. 

## 4. Discussion

The aims of the present study were two-fold: to identify behavioral traits, as rated by parents, and objective sleep patterns at the age of five years, to predict self-rated callous-unemotional traits and empathy at the age of 14, and to associate callous-unemotional traits and empathy with current dimensions of insomnia, stress, and mental toughness. The key findings of the present study were that higher scores for hyperactivity at age five significantly predicted higher callous-unemotional traits at age 14. Further, a lower score for negative peer relationships, that is, more favorable peer relationships, at age five significantly predicted higher scores for cognitive empathy at age 14. This pattern of results adds to the current literature in an important way, in that we were able to relate psychological traits at age five to callous-unemotional traits and empathy in adolescence. Lastly, a higher sleep efficiency and a shorter sleep latency, as assessed via objective sleep-EEG measures at the age of five years, predicted lower scores for callousness at the age of 14.

Six hypotheses were formulated, and each is considered now in turn.

With the first hypothesis, we expected that positive psychological traits in childhood would be associated with and even predict lower callous-unemotional traits and higher empathy scores at age 14, and the data confirmed these assumptions: lower hyperactivity scores predicted lower scores for callous-unemotional traits, and lower negative peer relationships predicted higher scores for empathy. As such, the present results confirmed previous data on callousness [53,62,63] and empathy [64,65,66]. The data, however, expand upon the current literature in that such data were observed among a sample of typically developing children over a period of nine years.

Next, based on a previous study [67], we expected that a higher sleep quality longitudinally (objective measure of sleep parameters at age five; Hypothesis 2) and lower scores for insomnia cross-sectionally (subjective measure of insomnia at age 14; Hypothesis 3) would be associated with lower callous-unemotional traits at age 14. Correlative computations showed that a higher objectively assessed sleep efficiency and a shorter sleep latency at age five were associated with lower scores for callousness at 14 years (confirmation of Hypothesis 2; though, we noted that the objective sleep dimension did not reach statistical significance in the regression model). Further, lower scores for insomnia were associated with lower scores for callous-unemotional traits (confirmation of Hypothesis 3).

Similarly, with the fourth and fifth hypotheses we predicted that a higher sleep quality longitudinally (objective measure of sleep parameters at age five; Hypothesis 4) would be associated with empathy at age 14. However, this hypothesis was rejected. In contrast, lower scores for insomnia cross-sectionally were associated with higher scores for cognitive empathy at age 14. As such, Hypothesis 5 was confirmed.

As regards the rejected fourth hypothesis, the quality of the data did not allow a deeper understanding of the underlying psychological mechanisms. Given this, the following four assumptions were made:

First, though highly speculative, it is conceivable that the study was underpowered. However, the longitudinal sample size of 77 should have allowed us to identify significant predictors. Second, data from developmental psychology [99,100] and sleep behavior [101] studies indicated that substantial changes in psychosocial and sleep patterns occur with the onset of adolescence. In this specific case, participants were 14 years old; thus, they had already crossed from childhood into adolescence. As a further consequence, it is conceivable that the psychological traits and sleep patterns had changed only within the last few years after starting adolescence. With this background in mind, it is conceivable that such changes could not be mirrored by longitudinal data starting at the preschool stage and developing in a linear fashion. Fourth, the longitudinal design was such to combine both objective and subjective sleep analysis; as such it is conceivable that these analyses did not mirror exactly the same dimensions of sleep. Indeed, there is evidence that objective sleep data and subjective sleep ratings may have a poor overlap [102,103,104].

There is growing evidence in the literature that sleep plays a critical role in emotional processing [105,106,107]. Tempesta et al. [108] were able to find consistent evidence in their extensive review that a lack of sleep significantly influences emotional reactivity. Rong et al. [109]. also suggested that sleep is important for the development of empathy and showed that a longer nightly sleep duration and fewer sleep disturbances are associated with a more mature empathy pattern in young preschool children.

With the sixth hypotheses, we expected that higher scores for callous-unemotional traits and lower scores for empathy would be associated with higher perceived stress and lower mental toughness traits, and the data generally confirmed these assumptions. Thus, the present data expanded upon the current literature by demonstrating that both callous-unemotional traits and low empathy were associated with high cognitive-emotional load (stress) and low resources for coping with cognitive-emotional load (mental toughness).

However, the present results did not shed any light on why sleep efficiency and positive psychological traits such as internalizing problems, hyperactivity, negative peer relationships, and prosocial behavior at 5 years predict callous-unemotional traits and empathy nine years later.

### 4.1. Behavioral Traits, Empathy, and Callous-Unemotional Traits

Previous studies have already shown that increased psychological problems in childhood also enhance the risk of increased psychological problems in adolescence and early adulthood [39,110,111], and that personality traits remain relatively stable from childhood to adolescence [111,112,113]. Furthermore, we also know that the expression of callous-unemotional traits in a large number of children decreases during childhood and adolescence, and that those who already show an increased level of callous-unemotional traits in childhood have an increased risk of maintaining them later in life [14].

### 4.2. Hyperactivity Predicts Callous-Unemotional Traits

To explain our finding that increased hyperactivity predicts higher callous-unemotional traits, we relied firstly on the study by Muratori et al. [63], who found that adolescents who showed elevated levels of hyperactivity also had higher callous-unemotional traits. Second, there is evidence that adolescents with conduct problems [114] and adolescents with a combination of callous-unemotional traits and conduct problems [115] are also likely to have increased levels of hyperactivity. Last, we relied on the fact that callous-unemotional traits, both in clinical and community samples, are closely related to externalizing behavior problems such as ADHD, ODD, and CD [14,116,117]. Since externalizing disorders include uncontrolled, impulsive, or aggressive behavior [118], the strong correlation of hyperactivity and callous-unemotional traits is coherent.

### 4.3. Peer Relationships Predict Empathy

The connection between peer relationships and empathy can be explained as follows: First, there is already evidence for a positive relationship between peer relationships [24,119,120,121] and empathy. Second, we relied on the findings of Boele et al. [122], who concluded in their meta-analysis that adolescents presenting a higher relationship quality with peers tend to care more about and understand the feelings of others. We suggest that positive peer relationships in early childhood underlie a learning process and thus promote further positive peer relationships, which in turn translate into better empathy. 

How should these findings be related to the present study? Our proposal is that higher levels of hyperactivity and worse peer relationships might be understood as a behavioral and emotional pattern that leads to antisocial behavior, which presents as increased callous-unemotional traits and a lack of empathy. In the present study, higher hyperactivity predicted more callous-unemotional traits 9 years later, and better peer relationships predicted more empathy, suggesting considerable stability in the level of psychological traits from childhood to mid-adolescence [71,111,123].

### 4.4. Sleep and Callous-Unemotional Traits

Previous research has shown that disturbed sleep or poor sleep quality is associated with various aspects of our functioning, including a range of psychiatric manifestations such as depression, anxiety, obsessive-compulsive disorder, and post-traumatic stress disorder [124,125]. However, researchers still are divided on the association between sleep and psychopathy, which includes callous-unemotional traits. While some researchers suggest a negative association [67] between callous-unemotional traits and sleep, other studies have shown a positive association [53,126].

Our longitudinal findings showed a correlation between objectively measured sleep efficiency and latency. However, no sleep variable at age five was a significant predictor of callous-unemotional traits at age 14, when objectively assessed sleep efficiency and sleep onset latency were entered into the regression model. The cross-section showed the same pattern of results: fewer sleep problems were associated with fewer callous-unemotional traits.

How do we explain our results, given the studies that suggest a positive correlation between poor sleep and higher scores for callous-unemotional traits? First, previous research has shown that callous-unemotional traits may moderate the relationship between sleep quality and externalizing problems, such that sleep problems occur only in adolescents with externalizing problems and low levels of callous-unemotional traits [53,54,125]. This association was not examined in our analysis. Second, only Backman et al. [67] examined adolescents. The other studies focused on adults, which further complicates the integration of our results into the current research. Third, there is no comparable longitudinal study with which we can compare our results. To summarize, we observed a correlative association between more favorable objective sleep at age five and lower scores for callousness at age 14. However, objective sleep parameters lost their predictive value, when entering both psychological dimensions such as hyperactivity and negative peer relationships and objective sleep parameters concomitantly into the regression model.

### 4.5. Limitations

Despite the novelty of the results, several limitations advise against their overgeneralization. First, the sample size was small. A larger sample would have had greater statistical power and might have revealed other significant relationships. Secondly, the pattern of results could have been due to other dimensions, such as genetic heritage, family functioning, and environmental influences, that were not recorded and which may have affected our results. In addition, parenting style, for example, was not recorded at both time points. In this regard, there is evidence that child and adolescent behavior and sleep are not independent of family life, parenting style [127], or parental sleep patterns [127,128,129]. Fourth, the objective measures of sleep (sleep-EEG) at the age of 5 were only valid for a subsample. It is possible that this subsample differed from the total sample and thus influenced the results. In addition, sleep at age 14 was assessed only subjectively. An objective assessment would have enhanced the quality of the data. Fifth, data on socioeconomic status and psychosocial risk factors were not included. This should be addressed in future studies.

## 5. Conclusions

Positive psychological traits as reflected in lower hyperactivity and lower negative peer relationships at age five were associated with lower callous-unemotional traits and more empathy at age 14. Cross-sectionally, callous-unemotional traits, poor sleep, and higher stress were associated.

Given this pattern of results, it appears plausible that parents, pediatricians, teachers, or even sports coaches should pay sufficient attention to children’s and adolescents’ sleep habits and sleep quality. This holds particularly true, as the present data showed that poor sleep during preschool years, as measured via portable sleep-EEG devices, might lead to unfavorable cognitive-emotional and behavioral issues during adolescence. In addition, poor subjectively assessed sleep during adolescence was associated with further mental health issues. To promote restoring sleep among children, parents and their children benefit from thorough sleep hygiene instructions, at least among 10-year-old children with ADHD [130]. To improve restoring sleep among adolescents, behavioral changes have proved to be particularly favorable [131,132,133]. Further, to treat problems of externalizing behavior as a proxy of callous-unemotional traits, social-learning-based parent training appears to be capable of producing lasting improvement in children’s callous-unemotional traits [134]. 

## Figures and Tables

**Table 1 jcm-12-02248-t001:** Descriptive statistical indices (means and standard deviations) and correlations between callous-unemotional traits and empathy at age 14 and parents’ ratings of children’s psychological functioning at age five.

	Dimensions at Age 14							
	Callous-Unemotional Traits (ICU)				Empathy (GEM)			Descriptive Statistics
	Callousness	Uncaring	Unemotional	Overall Score	Cognitive	Affective	Overall score	M (SD)
**Strengths and difficulties** (SDQ; parents’ ratings)								N = 77
Internalizing problems	−0.08	0.03	0.09	−0.01	−0.02	0.19	0.14	0.31 (0.36)
Externalizing problems	0.21 *	0.16	0.11	0.24 *	−0.23 *	0.04	−0.18	0.36 (0.27)
Negative peer relationships	0.06	0.01	0.05	0.06	−0.29 **	−0.04	−0.16	0.21 (0.29)
Hyperactivity	0.25 *	0.26 *	0.20 *	0.34 **	−0.16	0.16	0.02	0.52 (0.41)
Prosocial behavior	−0.13	−0.15	−0.07	−0.17	0.16	0.09	0.31 *	1.48 (0.39)
Overall score	0.18	0.20 *	0.19	0.27	−0.28 **	0.18	−0.04	0.35 (0.20)
**EEG**								N = 64
Sleep efficiency (%)	−0.42 *	−0.06	0.01	−0.22	−0.16	0.04	−0.03	94.63 (2.80)
Sleep latency (min)	0.37 *	0.01	−0.07	0.16	0.38 *	−0.04	0.14	12.05 (10.98)
Time in bed (min)	0.01	0.01	−0.01	0.01	−0.09	0.04	−0.04	641.31 (41.14)
Total sleep time (min)	−0.23	−0.01	0.01	−0.12	−0.18	0.06	−0.06	606.60 (38.38)

* *p* < 0.05; ** *p* < 0.01.

**Table 2 jcm-12-02248-t002:** Overview of the multiple regression analyses with callous-unemotional overall score and cognitive empathy scores at age 14 as dependent variables, and parents’ ratings of children’s strengths and difficulties (SDQ) and objectively assessed sleep efficiency at age five years as independent variables.

		Non-Standardized Coefficients		Standardized Coefficient						
Dimension	Variable	Coefficient Beta	Standard Error	beta	t	*p*	R	R^2^	R^2^ Corr	Durbin-Watson Statistics
Callous-unemotional traits	Intercept	20.510	4.540	-	4.518	<0.001	0.392	0.153	0.089	1.914
	Hyperactivity	5.356	2.150	0.308	2.492	0.015				
Variables not reaching statistical significance (=*p* > 0.05): internalizing problems; externalizing problems; negative peer relationships; prosocial behavior; objective sleep efficiency										
Empathy; cognitive traits	Intercept	14.020	4.270	-	3.283	0.002	0.394	0.153	0.124	2.073
	Negative peer relationships	−7.693	2.779	−0.326	−3.89	0.007				
Variables not reaching statistical significance (=*p* > 0.05): internalizing problems; externalizing problems; hyperactivity; prosocial behavior; objective sleep efficiency										

**Table 3 jcm-12-02248-t003:** Correlations between callous-unemotional traits and empathy and insomnia, perceived stress and mental toughness at age 14.

	Dimensions at Age 14							
	Callous-Unemotional Traits (ICU)				Empathy (GEM)			Descriptive Statistics
	Callousness	Uncaring	Unemotional	Overall Score	Cognitive	Affective	Overall Score	
Insomnia	0.29 **	0.17	0.24 *	0.34 **	−0.22 *	0.07	−0.09	5.65 (3.99)
Perceived stress	0.13	0.23 *	0.14	0.23 *	−0.12	0.14	0.04	4.26 (2.00)
Mental toughness	−0.05	−0.10	−0.21 *	−0.15	0.05	−0.10	−0.02	22.31 (4.85)

* *p* < 0.05; ** *p* < 0.01.

## Data Availability

Not applicable.

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
