# Peer review of "The Origins of the Dark—Hyperactivity and Negative Peer Relationships, an Objectively Lower Sleep Efficiency, and a Longer Sleep Onset Latency at Age Five Were Associated with Callous-Unemotional Traits and Low Empathy at Age 14"

_jcm, 2023, doi:10.3390/jcm12062248_

Round 1
Reviewer 1 Report
In the present work, Eisenhut et al. studied the association between different emotional and behavioral patterns along with objective sleep dimensions at the age of 5 years with emotional features in adolescents, defining an interesting and potential association between hyperactivity and peer relationship in the childhood with callous-emotional traits and low-empathy in teenagers.
Overall, the work is worthwhile, including a long period of follow-up, valuable results and a thorough search of the available literature. The results add valuable information to the field. However, I have some comments/suggestions that must be addressed to increase the quality of the manuscript.
1. Despite I find this topic really interesting, and I have enjoyed the reading, I feel that the structure of the manuscript should be improved, especially in the introduction and discussion part. Perhaps, the introduction can be shortened, and the discussion can be more focused on the results obtained or express the ideas in a more organized way.
2. Notwithstanding the studied sample are described in previous works, a summary table with the main sociodemographic characteristics of the children followed would be of aid to understand the studied sample.
3. Due to the emotional and behavioral differences between male and female, it is not clear to me why the gender is not considered in the statistical analysis.
4. Since the introduction part (1.3), there is a lack of uniformity in the references. Instead of numbering, it is designed by the name of the author (et al.) and the year. Please, unify the format according to the journal´s instructions.
5. Section 4.3 " Our proposal is that higher levels of hyperactivity and better peer-relationships might be understood as behavioral and emotional pattern that leads to antisocial behavior, which is featured by increased callous-unemotional traits and lack of empathy". Is this statement correct? Would not it be “worse peer relationships”?
Minor:
For facilitating further peer-review process, enumerating the lines of the work is strongly recommended.
Thus, both individual and environmental risk factors appeared to contributing to the development… à appear to contribute to the development
Author Response
In the present work, Eisenhut et al. studied the association between different emotional and behavioral patterns along with objective sleep dimensions at the age of 5 years with emotional features in adolescents, defining an interesting and potential association between hyperactivity and peer relationship in the childhood with callous-emotional traits and low-empathy in teenagers.
Authors' reply: We thank Reviewer #1 for their thorough and comprehensive summary of the present study and the present manuscript.
Overall, the work is worthwhile, including a long period of follow-up, valuable results and a thorough search of the available literature. The results add valuable information to the field. However, I have some comments/suggestions that must be addressed to increase the quality of the manuscript.
Authors' reply: We do highly appreciate the Reviewer’s statements, which encouraged and motivated us to further undertake efforts the resubmit a tidy and corrected revision.
- Despite I find this topic really interesting, and I have enjoyed the reading, I feel that the structure of the manuscript should be improved, especially in the introduction and discussion part. Perhaps, the introduction can be shortened, and the discussion can be more focused on the results obtained or express the ideas in a more organized way.
Authors' reply: Thank you for this valuable comment. As requested, we have slightly shortened the Introduction section and we have highlighted the subheadings; for the Discussion section, we have highlighted the subheadings, always with a blue background. Further, as requested, we have shortened some paragraphs; such shortenings were not reported in the present point-by-point-response; please see the full text.
2. Notwithstanding the studied sample are described in previous works, a summary table with the main sociodemographic characteristics of the children followed would be of aid to understand the studied sample.
Authors' reply: Thank you for this suggestion. As requested, we have provided more in information of the study sample, though, in a flow text and not in a table. The text reads:
“…In brief, preschool children from 18 different kindergartens in Basel (North-western and German-speaking part of Switzerland) were assessed. Those 18 kindergartens were randomly chosen, though considering a balanced stratification of different quartiers reflecting a broad range of the socioeconomic background, including the prevalence rates of immigrants with (Swiss-)German as the second language. Eligible preschoolers were.."
3. Due to the emotional and behavioral differences between male and female, it is not clear to me why the gender is not considered in the statistical analysis.
Authors' reply: Thank you for this candid question, which is entirely plausible, given that relative to males, females report more internalizing problems, and given that relative to females, males report more externalizing problems. However, for the following reasons, the decision was not to further investigating gender differences. From a statistical point of view, multiple testing would have led to more conservative p-corrections; in doing so, the odds of non-significant p-values would have increased. Relatedly, this would have led to the question of an underpowered study sample. Next, we have already formulated six hypotheses, and we think that any further hypothesis or research question would have rather diminished, that enriched the overall pattern of results. Again, we understand the scientific need to further exploring gender differences, though, for the present manuscript, the decision was not to further consider this research question.
4. Since the introduction part (1.3), there is a lack of uniformity in the references. Instead of numbering, it is designed by the name of the author (et al.) and the year. Please, unify the format according to the journal´s instructions.
Authors' reply: Thank you! We are working on this; as for now, for unknown reasons, the reference softwares (Zotero®; EndNote®) were always fully functional, and we are confident to solve this issue soon.
5. Section 4.3 " Our proposal is that higher levels of hyperactivity and better peer-relationships might be understood as behavioral and emotional pattern that leads to antisocial behavior, which is featured by increased callous-unemotional traits and lack of empathy". Is this statement correct? Would not it be “worse peer relationships”?
Authors' reply: We thank Reviewer #1 for their scrutiny, and we apologize for such a beginner’s mistake. The text reads as follows:
“….Our proposal is that higher levels of hyperactivity and worse better peer-relationships might be understood as behavioral and emotional pattern that leads to antisocial behavior…..”
Minor:
For facilitating further peer-review process, enumerating the lines of the work is strongly recommended.
Authors' reply: Thank you! As requested, we introduced the line numbering, though, sometimes, such features might get lost during the file transferring process.
Thus, both individual and environmental risk factors appeared to contributing to the development… à appear to contribute to the development
Authors' reply: Thank you: To shorten the Introduction section, this specific sentence has been deleted.
Authors' reply: Again, we thank Reviewer #1 for their statements and comments, which helped us to improve the quality of the present revision.
Reviewer 2 Report
REVIEW REPORT: The Origins of the Dark—Hyperactivity and Negative Peer
Relationships at Age Five Predict Callous-Unemotional Traits
and Low Empathy at Age 14
The authors have presented the results and findings of a longitudinal study on the associations between negative early childhood experiences and callous-unemotional traits in the late childhood period. The study has interesting results, and I appreciate the authors' contribution.
However, some major corrections need to be made before it can be considered for acceptance. Here are a few suggestions that would improve the manuscript.
· Please use any language editing software/service to correct some minor technical and grammar errors.
· Background (Introduction) is very lengthy; try summarising pages 2-4 into a paragraph or two, avoid too many references (67 references for just the introduction section is a little too much!!!), and give a brief background to the current study – including the scope and need of the study, what is already done in the area, and introduce the current study – objectives and research questions.
· Follow a consistent pattern for in-text citations (either numbered or author-date). Check the journal policies for the style of reference.
· Please specify what or where is ‘below’ in these sentences. (Strength and Difficulties Questionnaire (SDQ, see below); empathy (see below). The current usage is very ambiguous.
· Please include reliability and validity measures for the scales used. Also, provide references to the Cronbach alpha scores already included.
· Please include more information on the methodology – the study design and setting, participant recruitment, and data collection procedure. Authors can think about adding a flowchart of participant recruitment for more clarity. Also, specify the reasons for loss to follow up.
· In the results, please include brief information about the participants – the demographic variables.
· Please delete this sentence - All statistical indices are not repeated in the text again.
· Please specify somewhere that descriptive statistics refers to mean and standard deviation.
· You need not repeat the results in the discussion section. Just summarise major results and then discuss them in the context of other studies published. Implications for future practice, research, and policy can be discussed.
· Correct this - Data Availability Statement: In this section, please provide details regarding where data supporting reported results can be found, including links to publicly archived datasets analyzed or generated during the study. Please refer to the suggested Data Availability Statements in the section “MDPI Research Data Policies” at https://www.mdpi.com/ethics. If the study did not report any data, you
might add “Not applicable” here.

Author Response
The authors have presented the results and findings of a longitudinal study on the associations between negative early childhood experiences and callous-unemotional traits in the late childhood period. The study has interesting results, and I appreciate the authors' contribution.
However, some major corrections need to be made before it can be considered for acceptance. Here are a few suggestions that would improve the manuscript.
- Please use any language editing software/service to correct some minor technical and grammar errors.
Authors' reply: Thank you; also Reviewer #1 noted that grammar and style should be improved. As mentioned above, we did carefully edit the text once again for grammar and style. - Background (Introduction) is very lengthy; try summarising pages 2-4 into a paragraph or two, avoid too many references (67 references for just the introduction section is a little too much!!!), and give a brief background to the current study – including the scope and need of the study, what is already done in the area, and introduce the current study – objectives and research questions.
Authors' reply: Thank you for this candid comment. As requested, and as also Reviewer # 1 mentioned, we have slighty modified and trimmed the Introduction section. However, given the paucity of research in this field, the decision was to provide a thorough and comprehensive overview of the topics. - Follow a consistent pattern for in-text citations (either numbered or author-date). Check the journal policies for the style of reference.
Authors' reply: Thank you; it appears that we still have some issues to thoroughly merge the reference formats of Zotero® and EndNote®. As for now, sadly, we were unable to deal with this issue, and we apologize for such a beginner’s mistake. - Please specify what or where is ‘below’ in these sentences. (Strength and Difficulties Questionnaire (SDQ, see below); empathy (see below). The current usage is very ambiguous.
Authors' reply: Thank you; the text reads now:
“….and psychological functioning (Strength and Difficulties Questionnaire (SDQ, see description of the measure below: 2.3.1.1. Preschoolers’ strengths and difficulties—parent rating) as assessed via parents’ and teachers’ ratings. Approximately 9 years later, 84 (88.43%) of the…”
“…self-assessment questionnaires on socio-demographic data, sleep, callous-unemotional behavior and empathy (see description below: 2.3.2.1. Callous-unemotional traits). The adolescents….”
“….parent ratings and EEG data; see description below: 2.3.1.2. Preschoolers’ objective sleep patterns) and….”
“….and empathy (ICU and GEM; see below: 2.3.2.1. Callous-unemotional traits; 2.3.2.2. Empathy). This pattern of…” - Please include reliability and validity measures for the scales used. Also, provide references to the Cronbach alpha scores already included.
Authors' reply: Thank you! Also Reviewer #3 asked for more psychometric details of the measures; such information were added to the text (and see also the statements for Reviewer #3. - Please include more information on the methodology – the study design and setting, participant recruitment, and data collection procedure. Authors can think about adding a flowchart of participant recruitment for more clarity. Also, specify the reasons for loss to follow up.
Authors' reply: Thank you! Also Reviewer #1 asked for more details.
The text reads now:
“…In brief, preschool children from 18 different kindergartens in Basel (North-western and German-speaking part of Switzerland) were assessed. Those 18 kindergartens were randomly chosen, though considering a balanced stratification of different quartiers reflecting a broad range of the socioeconomic background, including the prevalence rates of immigrants with (Swiss-)German as the second language. Eligible preschoolers were..”
As regards the reasons for loss to follow-up, sadly, adolescents and their parents, who did not answer to the follow-up inivitation, neither provided specific reasons. However, in our opinion, it was important that those who followed the invitation and those who did not follow the invitation did not systematically differ in their baseline values. This observation has been specified as follows:
As already shown before [4], participants and non-participants did not differ significantly at follow-up with regard to age, gender, sleep profiles or psychological traits at baseline. Accordingly, age, gender, sleep profiles and psychological traits were not included as covariates in all statistical equations of the present study.
- In the results, please include brief information about the participants – the demographic variables.
Authors' reply: Thank you. The text reads now:
3.0. Sociodemographic information
At follow-up, the mean age of the adolescents was 14.25 years (SD =1.21; 32 females and 45 males). Please note that while cross-sectional data were available of 84 partici-pants, longitudinal data were available of 77 participants.
- Please delete this sentence - All statistical indices are not repeated in the text again.
Authors' reply: As requested, this sentence has been deleted. - Please specify somewhere that descriptive statistics refers to mean and standard deviation.
Authors' reply: Yes, of course; we apologize for such an elementary mistake.
The title of Table 1 reads now:
Table 1. Descriptive statistical indices (means and standard deviations) and correlations between callous-unemotional traits and empathy at age 14 and parents’ rating of children’s psychological functioning at age five.
Further, the title of the last column of Table 1 reporting the descriptive values has been highlighted in blue. - You need not repeat the results in the discussion section. Just summarise major results and then discuss them in the context of other studies published. Implications for future practice, research, and policy can be discussed.
Authors' reply: Thank you. The set-up of the Discussion section is often a question of taste and judgement; as for now, we made good experiences and we got encouraging feedbacks from Reviewers and Editorial Board with such a kind of Discussion.
Besides, we hightly appreciated the suggestion to report Implications for future practice, research, and policy.
The text reads now:
Given this pattern of results, it appears plausible that parents, pediatricians, teachers, or even sports coaches should pay sufficient attention to children’s and adolescents’ sleep habits and sleep quality. This holds particularly true, as the present data showed that poor sleep during preschool years, as measured via portalble sleep-EEG devices, might lead to unfavorable cognitive-emotional and behavioral issue during adolescence. In addition, poor subjectively assessed sleep during adolescence was associated with further mental health issues. To promote restoring sleep among children, parents and their children benefit from thorough sleep hygiene instructions, at least among 10 years old children with ADHD {Keshavarzi, 2014 #1220}. To improve restoring sleep among adolescents, behavioral changes proved to be particularly favorable {Das-Friebel, 2019 #5158;Dewald-Kaufmann, 2014 #2948;Dewald-Kaufmann, 2013 #5409). Further, to treat problems of externalizing behavior as a proxy of callous-unemotional traits, social-learning-based parent training appears to be capable of producing lasting improvement in children’s callous-unemotional traits {Hawes, 2014 #7151}. - Correct this - Data Availability Statement: In this section, please provide details regarding where data supporting reported results can be found, including links to publicly archived datasets analyzed or generated during the study. Please refer to the suggested Data Availability Statements in the section “MDPI Research Data Policies” at https://www.mdpi.com/ethics. If the study did not report any data, you might add “Not applicable” here.
Authors' reply: Yes, correct! Thank you!
Authors' reply: Again, we thank Reviewer #2 for their statements and comments, which helped us to improve the quality of the present revision.
Reviewer 3 Report
1. The title is very interesting and has the academic value of this topic.
2. In the section of the introduction, the author well organized more literature review and provided more important and critical viewpoints on this topic.
3. In the section of the measurement, I suggest the author could provide some literature review or backgrounds to support the tool’s rationales.
4. In the section of the results, I suggest the author could provide more information of the validity and reliability of employed measures or scales in this study.
Author Response
- The title is very interesting and has the academic value of this topic.
Authors' reply: Thank you so much for such encouraging comments. Please note that we did slightly modify the title such to providing a more accurate summary of the pattern of results. The title reads now:
The Origins of the Dark—Hyperactivity and Negative Peer Relationships, an Objectively Lower Sleep Efficiency and a Longer Sleep Onset Latency at Age Five were associated with Callous-Unemotional Traits and Low Empathy at Age 14
To evidence that modifications of the title, the following paragraph was introduced in the Discussion section:
To summarize, we observed a correlative association between more favorable objective sleep at age five and lower scores for callousness at age 14. However, objective sleep parameters lost their predictive value, when entering both psychological dimensions such as hyperactivity and negative peer-relationship and objective sleep parameters concomitantly in the regression model.
- In the section of the introduction, the author well organized more literature review and provided more important and critical viewpoints on this topic.
Authors' reply: Again, we thank Reviewer #3 for their positive and motivating statements; we did highly appreciate this. - In the section of the measurement, I suggest the author could provide some literature review or backgrounds to support the tool’s rationales.
Authors' reply: Thank you! We introduced the following information:
- Measures applied at the age of five (SDQ, chapter 2.3.1.1 à line 287):
-New sentence at the end of the section: The SDQ is considered a valid behavioral screening tool in RCTs and clinical settings (Hall et al. 2019), both in children and adolescents (Mieloo et al. 2012; Goodman 2001) à line 296-2997
- Callous-unemotional traits (ICU, chapter 2.3.2.1 à line 310)
-The ICU is the most commonly used measure of callous unemotional traits and is used as a self-report and observational report version (Cardinale and Marsh 2020; Deng et al. 2019) à line 312-313
-New sentence at the end of the section: The ICU total score (Frick et al. 2014; Frick and Ray 2015), as well as the callous and uncaring subscales (Cardinale and Marsh 2020) are considered to be a valid measure for callous-unemotional traits in children adolescents. However, Cardinale and Marsh (2020) also point out the weakness of the unemotional subscale as it relates to interpersonal callousness, uncaring, and antisociality à line 318-323
- Empathy (GEM, chapter 2.3.2.2 à line 324)
-New sentence at the end of the section: The GEM can be used in children and adolescents to measure empathic behavior and showed good reliability and validity across gender and age (Dadds et al. 2008). à line 330-332
- Insomnia (ISS, chapter 2.3.2.3 à line 333)
- The 7-item questionnaire is a comprehensive screening measure for insomnia à 335-336
- The validity and reliability of the ISI has been proven previously (Fernandez-Mendoza et al., 2012; Gerber et al., 2016) à line 339-340
à no new sentences were added
- Perceived stress (PSS, chapter 2.3.2.4 à line 343)
-New sentence: Various studies have shown that the PSS is a valid instrument for assessing perceived stress in different languages, cultures and age groups (Baik et al. 2019; Chen et al. 2021; Ezzati et al. 2014; Lee and Jeong 2019) à line 349-352
- Mental toughness (MTQ, chapter 2.3.2.5 àline 361)
- In the section of the results, I suggest the author could provide more information of the validity and reliability of employed measures or scales in this study.
Authors' reply: Thank you; we have provided the specific Cronbach’s alphas at the end of the description of each measure.
Authors' reply: Again, we thank Reviewer #3 for their statements and comments, which helped us to improve the quality of the present revision.
Round 2
Reviewer 1 Report
The authors have responded satisfactorily to all the comments
Reviewer 2 Report
Dear Authors,
Thank you for reworking the manuscript to make it sound. All the very best